# EGOCENTRIC VISION LANGUAGE PLANNING

## ABSTRACT

We explore leveraging large multi-modal models (LMMs) and Text2image models
to build a more general embodied agent. LMMs excel in planning long-horizon
tasks over symbolic abstractions but struggle with grounding in the physical world,
often failing to accurately identify object positions in images. A bridge is needed
to connect LMMs to the physical world. The paper proposes a novel approach,
egocentric vision language planning (EgoPlan), to handle long-horizon tasks from
an egocentric perspective in varying household scenarios. This pipeline leverages a
diffusion model to simulate the fundamental dynamics between states and actions,
discusses how to integrate computer vision related techniques like style transfer and
optical flow to enhance ability of modeling spatial states and generalization across
different environmental dynamics. The LMM serves as a planner, breaking down
instructions into sub-goals and selecting actions based on their alignment with
these sub-goals, thus enabling more generalized and effective decision-making.
By using LMM, we can output text actions, using a series of mechanisms such as
reflection to perform high-level task decomposition and low-level action output
end-to-end. Experiments show that EgoPlan improves long-horizon task success
rates from the egocentric view compared to baselines across household scenarios.

## 1 INTRODUCTION

The advent of large language models (LLMs) (et al., 2024b; Touvron et al., 2023) and large multi-
modal models (LMMs) (202, 2023; Girdhar et al., 2023; Zhang et al., 2023a; Zhu et al., 2023) has
revolutionized the field of artificial intelligence. Their strong reasoning (Wang et al., 2023b; Wei et al.,
2023) and powerful generalization capabilities allow them to be directly applied in various scenarios.
In the next step toward artificial general intelligence (AGI), researchers are considering enabling
large models (LMs), especially LMMs, to break through the world expressed by text and images
to interact with the physical world. They aim to build a general embodied agent that intelligently
interacts with the physical world.

LMMs have demonstrated an impressive capability of planning for long-horizon tasks over symbolic
abstraction in the physical world (Wake et al., 2024). However, there's still a piece of the puzzle
missing. They have struggled to ground the text world with the physical world. For example, GPT-4V
often fails to accurately identify objects' positions in images. LMMs seem to know *what to do next*
but do not understand *how the world works*. A world model (dynamics model) is hence needed to
connect the LMMs to the physical world. There are two potential solutions. One is to implicitly
integrate environmental dynamics into the LMMs, that is, fine-tuning the LMMs based on a vast
amount of state-action sequences, such as PaLM-E (Driess et al., 2023) and RT-2 (et al., 2023).
However, directly training large models requires extensive data and computational resources. The
other is to explicitly introduce a pre-trained world model, *e.g.,* Text2image models (Radford et al.,
2021; Saharia et al., 2022), which can be used by LMMs as an auxiliary tool. Our work explores the
second path. We try to answer the question: ***can we leverage the LMMs and Text2image model to
build a more general embodied agent?***

Some works already train Text2image/video models as world models for decision-making. However,
there still exist several limitations. First, their task scenarios often involve object manipulation, a fully
observable setting. This is uncommon in real-world scenarios, and their methods seem to struggle to
adapt to other practical scenes. For example, SuSIE (Black et al., 2023) and VLP (Du et al., 2023b)
require generating images several steps ahead, yet the error introduced by long-range predictions is
substantial for most partially observed scenarios, *e.g.,* autonomous driving. In contrast, we focus

on a more challenging, partially observable setting. The embodied agent, like humans, tends to complete more complex and comprehensive tasks, *e.g.,* household tasks including manipulation and navigation, from the egocentric view. Second, their framework has limited capability, mainly reflected in three aspects: (i) their low-level policies are tailored to specific tasks, and generalize the polices to new dynamics can lead to policy collapse; (ii) a key challenge in world models is how to represent the mapping between state transitions and action information. In the aforementioned work, action information is typically represented in text form. However, this representation is too coarse, making it difficult to establish a mapping between coarse-grained text actions and fine-grained state transitions, especially in comprehensive tasks and partially observable environments. (iii) The dynamics can vary for different entities even given the same action described by the text, *e.g.,* turn left, due to the inherent different in these entities. The Text2image/Text2video world model lacks individual motion pattern information and cannot be generalized accurately to dynamics of other environments that are out of the training dataset. We hope the agent can generalize to different dynamics within the fixed household scenario.

In this work, we propose egocentric vision language planning (**EgoPlan**), a general embodied agent to perform long-horizon tasks from the egocentric view and slove these three questions mentioned above. In our approach, we draw on perspectives and methods from the field of computer vision to enhance the world model. In a range of studies, optical flow is frequently utilized for human/robot action prediction (Ko et al., 2023) and scene understanding (Yang & Ramanan, 2020). This underscores the rich information regarding actions and state transitions contained within optical flow data. Compared to traditional text-based actions in world models, integrating optical flow into these models for task planning could enhance spatial orientation understanding in navigation tasks and facilitate the modeling of object motion prediction in manipulation tasks. Style transfer in computer vision enables the integration of diverse content semantics and fine-grained image styles using a limited number of samples. This capability can significantly enhance the world model's ability to perform fine-grained texture modeling and generation across different scenes.

We conduct a comprehensive evaluation and analysis of each module of the embodied agent. Empirically, we demonstrate the high quality of image generation by the world model and the high accuracy of optical flow prediction. Subsequently, we verify the world model's effectiveness in aiding decision-making in more complex tasks. Lastly, we confirm the method's generalization capabilities in a different environment. Our major contributions are summarized as follows:

- We have collected a dataset on Virtualhome, which views an action of the agent as a trajectory and provides egocentric observations each time-step and fine-grained action information, visualising optical flow, depth maps and semantic segmentation maps at each time step in the trajectory, which will provide data support for navigation and manipulation tasks in the embodied environment.

- We propose **EgoPlan**, a framework for complex task planning that combines LMM and a world model that predicts an egocentric view of the scene at the next time step after an action is executed and the scene of the subgoal is completed. Optical flow information is computationally invariant to different scenes and styles motion. Introduce optical flow into the world model leads the world model more sensitive to action position changes and adapt to scene changes during navigation. Then we borrow the idea of style transfer in computer vision and adopt the LoRA (Hu et al., 2021) model to fine-tune our diffusion world model by a small number of sample images, so as to enhance the ability of our framework to achieve few-shot generalization in different embodied scenarios.

- For the action selection and decision-making module, we employ the LMM as the execution module in both the high-level task decomposition and low-level action selection components. The LMM's strong multimodal understanding, reasoning capabilities, and text output abilities enable us to utilize a series of reflection and summarization mechanisms to accomplish tasks, while also ensuring the agent inherits this ability of generalizing the downstream polices to new dynamics. We demonstrate the effectiveness of our framework through LMM+world model planning experiments on comprehensive tasks.

## 2 RELATED WORK

In this section, we present a brief overview of related work. More discussions are in Appendix A.

## 2.1 DIFFUSION MODEL

The diffusion model (Ho et al., 2020; Song et al., 2022) has been extensively studied in the field of image generation (Dhariwal & Nichol, 2021; Ho et al., 2021; Rombach et al., 2022) and image editing (Gal et al., 2022; Hertz et al., 2022; Meng et al., 2022). Diffusion models can achieve a high degree of control during the image generation. In more detail, InstructPix2Pix (InstructP2P) (Brooks et al., 2023) trains a conditional diffusion model that, given an input image and text instruction for how to edit it, generates the edited image. ControlNet (Zhang et al., 2023b) is widely used to control the style of the generated image by using various forms of prior information, *e.g.*, edge information and segmentation. By adding LoRA or adapter (Houlsby et al., 2019) modules to the network, the model trained on one data distribution can also be transferred to other data distributions (different visual styles) through a few picture examples. The images produced by current diffusion models are of very high quality, highly realistic, and easily controllable. It prompts various fields to consider using these generated images to assist in accomplishing other tasks. Our paper adopts the diffusion model to generate task subgoals and predict the image of the next state for decision-making.

## 2.2 WORLD MODEL FOR DECISION-MAKING

The world model is used to model the dynamics of the environment. It is crucial for building autonomous agents and enabling intelligent interactions in various scenarios. However, developing a precise world model remains a significant challenge in model-based decision-making. The advancements in diffusion-based world models are reshaping how we model physical motion laws in real-world settings, particularly in robotics. UniPi (Du et al., 2023a) frames the decision-making problem in robotics as a Text2video task. The generated video is fed into an inverse dynamics model (IDM) that extracts underlying low-level control actions, which are executed in simulation or by a real robot agent. Video Language Planning (VLP) (Du et al., 2023b) introduces a novel method for task planning that integrates video generation with tree search algorithms. This methodology lets robots plan over longer horizons by visualizing future actions and outcomes. Unlike previous works, SuSIE (Black et al., 2023) leverages pre-trained image-editing models to predict the hypothetical future frame. A low-level goal-reaching policy is trained on robot data to reach this hypothetical future frame. Since one goal frame prediction does not require the model to understand the intricacies of the robot's low-level precisely dynamics, it should facilitate transfer from other data sources, *e.g.*, human videos. RoboDreamer (Zhou et al., 2024) advances the field by utilizing video diffusion to formulate plans combining actions and objects, solving novel tasks in unexplored robotic environments. We find it unrealistic to apply the Text2video model to partially observed scenarios. Moreover, it is still hard to predict the goal frame several steps ahead, as the shift in perspective could be significant. Therefore, we adopt the Text2image model to accurately predict the short-range outcome for one-step planning.

## 3 VH-1.5M DATASET

Most datasets related to embodied agents, *e.g.*, RT-X (et al., 2024a) and RH20T (Fang et al., 2023), employ the third-person view to avoid the visual occlusion issue, thus lacking data regarding the egocentric view (first-person view). There are some datasets, *e.g.*, Alfred (Shridhar et al., 2020) and Procthor (Deitke et al., 2022), that adopt a first-person perspective, however, they simplify the state transition by assuming instantaneous completion of actions, which fails to mimic the dynamics changes in real-world environments. We propose the VH-1.5M dataset based on the VirtualHome (Puig et al., 2018; 2020) environment to address these limitations.

We construct our dataset VH-1.5M in the VirtualHome environment, which comprises 50 distinct houses. Each house contains approximately 300 interactive objects, and the embodied agent can perform more than 10 actions. Note that the VirtualHome environment is a simulator tailored for embodied agents, offering a detailed simulation of a residential living scenario. It enables a range of household tasks, *e.g.*, navigation and object manipulation.

The VH-1.5M dataset is organized in a structured manner, encapsulating the relationship between actions, houses, agents, and trajectories. Each task sequence entry follows a hierarchical structure, *e.g.*, "/open/house_0/Female4/2_fridge" (female4 open the fridge2 in house0).

**Dataset Details:** The VH-1.5M dataset consists of:

| (a) observation | (b) next observation | (c) seg_inst | (d) depth | (e) optical flow |

Figure 1: An illustration sample in VH-1.5M, which includes current image observation, next image observation given the text action, semantic segmentation map, depth map, and optical flow map.

- 13 Actions: Various physical actions and interactions for agents within the houses.
- 50 Houses: Uniquely designed houses with diverse layouts and object placements.
- 4 Agents: Four distinct agents, each capable of performing the full range of actions.
- 1.5M Samples: Dateset has numerous detailed sequences, each executing one action. Information from each step in the sequence is stored as one sample. One example is shown in Figure 1. We use *House49* as the validation set.

More details of the dataset can be found in the Appendix C, and ***we will open-source the dataset.***

## 4 METHOD

Our embodied agent, EgoPlan, takes visual observation $x_t$ of the scene at the current timestep $t$ and a natural language goal $g$ as inputs and outputs an action $a_t$ to interact with the environment. Note that the $x_t$ only partially represents the current environment state. In addition, the agent uses encapsulated skills as actions, such as moving forward, turning, and grabbing objects.

EgoPlan consists of two parts, as illustrated in Figure 2. The first is a dynamics model that gives the agent the concept of the current environment, and the other is the planner that endows the agent with decision-making capabilities. Intuitively, we humans first envision the outcomes of each action in our minds, and then, by comparing the results, we make the best decision. In the same way, we use a dynamic model to create an egocentric scenario where different actions can be taken, which is then fed into LMM to determine which action is more reasonable.

### 4.1 DIFFUSION-BASED DYNAMICS MODEL

#### 4.1.1 LEARNING DYNAMICS

From a first-person perspective, the view after two or more steps may be completely different, making it difficult to model. Therefore, we aim to model the fundamental dynamics model, $p_\theta(x_{t+1}|x_t, a_t)$, for one-step planning usage. In more detail, we want to generate a new image $x_{t+1}$, representing the next state given the current visual observation $x_t$ and the text of the action $a_t$. Then, we cast our eyes on the Text2image model and resort to the diffusion model for modeling specifically. It has an irreplaceable advantage in easily incorporating other modalities as a condition.

Although the open-sourced diffusion model (Ho et al., 2022; Luo et al., 2023), $p_\theta(x_{\text{tar}}|x_{\text{src}}, l)$, trained on a wealth of online videos, has demonstrated the ability to predict the future, their generated results are hard to control, and most are only semantically reasonable. Moreover, most of the text in the pre-trained dataset consists of image descriptions $l$ rather than action instructions $a$. Therefore, supervised fine-tuning is adopted based on our VH-1.5M dataset to better model the dynamics, $p_{\theta_{\text{sft}}}(x_{t+1}|x_t, a_t)$. Formally, the training objective is given by:

$$\mathcal{L}_{\text{MSE}} = \left\| \epsilon - \epsilon_\theta \left( q \left( x_{t+1}^{(k)} | x_t, a_t \right), k \right) \right\|^2 \tag{1}$$

$$= \left\| \epsilon - \epsilon_\theta \left( \sqrt{\overline{\alpha_t}} x_t + \sqrt{1 - \overline{\alpha_t}} \epsilon | a_t \right) \right\|^2 \tag{2}$$

where $\epsilon_\theta$ is a learnable denoising model for reverse process, $k$ is denoising steps, and $\overline{\alpha_t}$ are a set of $K$ different noise levels for each $k \in [1, K]$, and $x_t$, $a_t$ separately represent the current observation image and action description text. However, we find it difficult to generalize directly to other environments since our dataset only includes VirtualHome scenes. The difference between two environments, *e.g.,* Habitat 2.0 (Savva et al., 2019; Szot et al., 2022) and VirtualHome, primarily lies in their different motion patterns for the same action and distinct visual styles. Especially for the

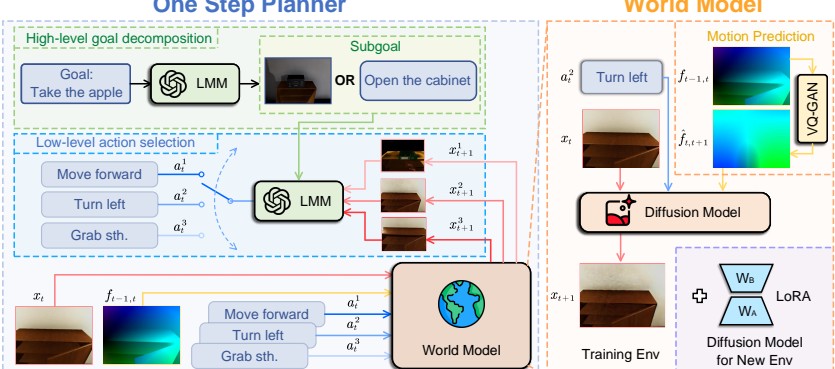

Figure 2: Overview of EgoPlan. The left side features a one-step planner that provides the agent with decision-making capabilities, while the right side includes a world model (dynamics model) that provides the agent with an understanding of the current environment.

former, the motion pattern, *e.g.,* the amplitude of the same action, performed by agents in a different environment can be unpredictable.

### 4.1.2 GENERALIZATION

We want to improve the model's generalization ability from a different perspective. In other words, instead of enhancing generalization through big data and large models, we aim to explicitly address the differences between environments such as the visual style of indoor environments and the definition of action amplitudes at the methodological level.

**Motion Regularization.** Firstly, we must combine the motion information into the diffusion model to distinguish the different motion patterns. Optical flow has thus caught our attention. It refers to the pattern of apparent motion of image objects between two consecutive frames caused by objects or camera movement. In optical flow maps, colors represent the direction of motion, and the depth or intensity of the colors indicates the magnitude of the motion, which is a general feature across different environments.

However, in practice, in the absence of the next observation, we cannot obtain the current optical flow, $f_{t,t+1}$. Inspired by other motion estimation works (Chen & Koltun, 2016; Zach et al., 2007), we assume motion consistency holds over short intervals, meaning abrupt changes do not occur. Consequently, the consecutive optical flow maps are highly correlated, allowing us to predict the current optical flow map using the previous map. The previous map is calculated from the previous two frames and reflects the actual motion pattern in the current environment.

We notice that optical flow generation does not require complex texture generation, and it is expected not to cause a significant delay in the pipeline. Therefore, we adopt a less powerful but lightweight generative model, VQ-GAN (Esser et al., 2021), and train it on our dataset to predict the optical flow map. Empirically, the generalization ability to predict optical flow is much better than predicting actual images. Formally, the training objective is given by:

$$\min \mathcal{L}_{VQ}(E, G, Z) = \|x - \hat{x}\|_2^2 + \|\text{sg}[E(x)] - z_q\|_2^2 + \beta\|\text{sg}[z_q] - E(x)\|_2^2, \tag{3}$$

where $E$ is the encoder, $G$ is the generator, $Z$ represents the latent space, $x$ is the input image, $\hat{x}$ is the reconstructed image, $z_q$ is the quantized latent vector, sg denotes the stop-gradient operator, and $\beta$ is a hyperparameter that balances the commitment loss.

*In summary, we use a simple model to predict motion patterns and then a more complex model to reconstruct real textures based on motion patterns.* Therefore, we adopt ControlNet (Zhang et al., 2023b) to incorporate the optical flow map, $f_{t,t+1}$, into the default diffusion model, $p_{\theta_{\text{sft}}}(x_{t+1}|x_t, a_t, f_{t,t+1})$. Only the ControlNet part needs to be fine-tuned on VH-1.5M at this stage. Formally, the training objective is given by:

$$\mathcal{L}_{\text{MSE}} = \left\| \epsilon - \epsilon_\theta \left( q \left( x_{t+1}^{(k)} | x_t, a_t, f_{t,t+1} \right), k \right) \right\|^2 \tag{4}$$

$$= \left\| \epsilon - \epsilon_\theta \left( \sqrt{\overline{\alpha_t}} x_t + \sqrt{1 - \overline{\alpha_t}} \epsilon | a_t, f_{t,t+1} \right) \right\|^2. \tag{5}$$

**Style Transfer.** Secondly, we use LoRA to fine-tune the diffusion model for visual style transfer. Note that LoRA requires very little data, just about 20 of samples. Normally, it is convenient to

collect data on such a scale in new environments. We expect the model to achieve generalization with as little effort as possible. In Section 5.2, we can find the role of LoRA method in maintaining the action pattern of the model between different environments, while flexibly transferring the style of fine-grained observation images.

## 4.2 PLANNING WITH DYNAMICS MODEL

To avoid further training in new environments, we prompt the LMM, *i.e.,* GPT-4V, as the planner. The LMM needs to be responsible for high-level goal decomposition as well as low-level action selection. Meanwhile, the pre-trained dynamics model can help the LMM better understand the world.

### 4.2.1 GOAL DECOMPOSITION

For long-term complex tasks, goal decomposition is an indispensable step. Subgoals can be represented in both text and image forms. For the text-based subgoal $g_{\text{tar}}$, we prompt the LMM to generate a reasonable one. In addition, we train another diffusion model, $p_{\theta_{\text{sft}}}(x_{\text{tar}}|x_t, g_{\text{tar}})$, to generate the image-based subgoal $x_{\text{tar}}$ only based on the text-based subgoal and current observation. Note that in order to complete long-horizon planning, the diffusion model is used in series of works to predict the scene image of the state when the subgoal task is completed (Black et al., 2023; Zhou et al., 2024), but these works mainly focus on manipulation task. For composite tasks that integrate manipulation and navigation, especially for navigation tasks, it is often quite difficult to generate subgoal scene images, because the subgoal scene images often involve the change of the entire image scene information, and the joint position of most objects changes, which requires the model's ability to understand spatial attributes. Not just editing the part of the image that involves an item. So predicting the image of the subgoal can be more challenging than predicting the next observation, which means the results are not very precise. We plan to delve into the impact of different types of subgoals on tasks. See Section 5.4.

### 4.2.2 ONE-STEP PLANNER

Since we can only ensure that the prediction for the next step is relatively accurate, we adopt a one-step planning method. In more detail, we utilize the pre-trained dynamics model to predict the visual outcomes of all the actions in the next state. Once the text/image-based subgoal is obtained, we send the subgoal and all the visual outcomes to the LMM. Then, we prompt it to compare all the potential outcomes with the subgoal and determine which action can bring the agent closer to the goal. So the process of goal decomposition and one-step planner is equivalent to the following formula.

$$\{G_0, G_1, \cdots, G_n\} = LLM(s_0, task) \tag{6}$$

$$a^* = \arg\min_{a \in A} d\left(f(s_t, a), G \in \{G_0, G_1, \cdots, G_n\}\right) \tag{7}$$

In the aforementioned equations, $\{G_0, G_1, \cdots, G_n\}$ refers to a series of subgoals that are decomposed from the task using LMM. It is noteworthy that, in selecting the optimal action for one-step planning process, inspired by Tan et al. (2024); Zhai et al. (2024), we utilize LMM to generate low-level actions in contrast to reinforcement learning or imitation learning algorithms. In this context, we leverage the comprehension capabilities of LMM to ensure the generalization of the low-level action in cross-environment decision-making. We also employing mechanisms like React (Yao et al., 2023) and Reflexion (Shinn et al., 2023) to enhance the agent's performance, which are shown in Appendix G. The prompt of task-decomposition and low-level action selection has been listed in Appendix F. Black et al. (2023) has discussed the generalization of objects concerning various operational targets; however, the generalization of underlying policy networks based on reinforcement learning or imitation learning algorithms, particularly in response to changes in the entire environmental scene—especially in navigation tasks, the ability of the pipeline still requires improvement. We will further discuss the experimental outcomes related to this in Sections 5.2 and 5.4.

## 5 EXPERIMENT

In this section, we comprehensively evaluate and analyze each module of the embodied agent. We first evaluate the quality of image generation using the world model and the quality of optical flow

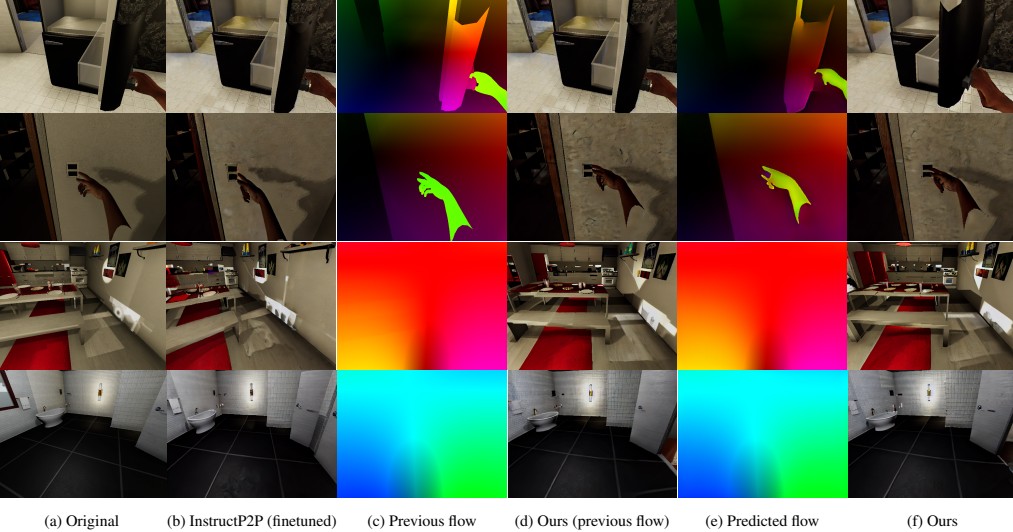

| (a) Original | (b) InstructP2P (finetuned) | (c) Previous flow | (d) Ours (previous flow) | (e) Predicted flow | (f) Ours |

Figure 3: Examples of the generated image of the next observation in VirtualHome. The tasks from rows 1 to 4 are: close the fridge, switch off the light, turn left, and turn right.

prediction. Secondly, we evaluate whether our world model can assist task planners in completing more complex tasks. Finally, we assess the generalization of our method.

## 5.1 VISUAL QUALITY

We adopt two metrics, FID (Heusel et al., 2018) and user score, to evaluate the visual quality of the generated image of the world model. For models, **InstructP2P (pre-trained)** is the default model of InstructP2P. **InstructP2P (fine-tuned)** is the model fine-tuned on our dataset. **Ours (previous flow)** is the world model that conditions on the previous optical flow map, while **Ours** is conditioned on the predicted optical flow map. Note that the validation set of VH-1.5M has around 5k samples.

Table 1: FID score comparison with other models on the validation set. It is calculated between the predicted observation and ground truth. The lower the number, the better the quality of the image.

| Model | Mean | Variance |
|---|---|---|
| InstructP2P (pre-trained) | 13.65 | 0.10 |
| InstructP2P (fine-tuned) | 1.06 | 0.05 |
| Ours (previous flow) | 0.83 | 0.03 |
| Ours | **0.82** | 0.03 |

**FID Score.** FID is a standard metric measuring the distance of two image distributions using the inception model. The smaller the FID is, the more similar the two images are. Table 1 shows the FID score of our model and baselines. We can see that using existing diffusion models as world models is ineffective because their training data often lacks state transition-related data. Meanwhile, introducing an optical flow map, which serves as motion pattern information, significantly enhances the generation results. In addition, world models based on predicted optical flow are slightly better than those based on the optical flow of the previous frame.

**User Study.** We also conduct a user study on the accuracy of world models for image generation. For the criterion, users judge the correctness of the direction and amplitude of the executed action. Each user investigates a total of 1000 samples from the validation set. There are 8 users participating in the survey in total. Our user study, shown in Table 2, again verifies our predicted optical flow can help generate higher-quality images.

Table 2: User score of the user study. The user score is the percentage of images that users consider to meet the criteria out of the total 1000 images. The higher the number, the better the quality of the image.

| Model | Mean | Variance |
|---|---|---|
| InstructP2P (fine-tuned) | 54.10% | 1.53% |
| Ours (previous flow) | 69.35% | 1.34% |
| Ours | **74.93**% | 2.57% |

**Analysis.** As illustrated in Figure 3, InstructP2P (fine-tuned) generates the scene of steering in the wrong direction. However, this flaw can be greatly improved by incorporating optical flow information. Moreover, it is observed that the dynamics of closing the refrigerator can be more

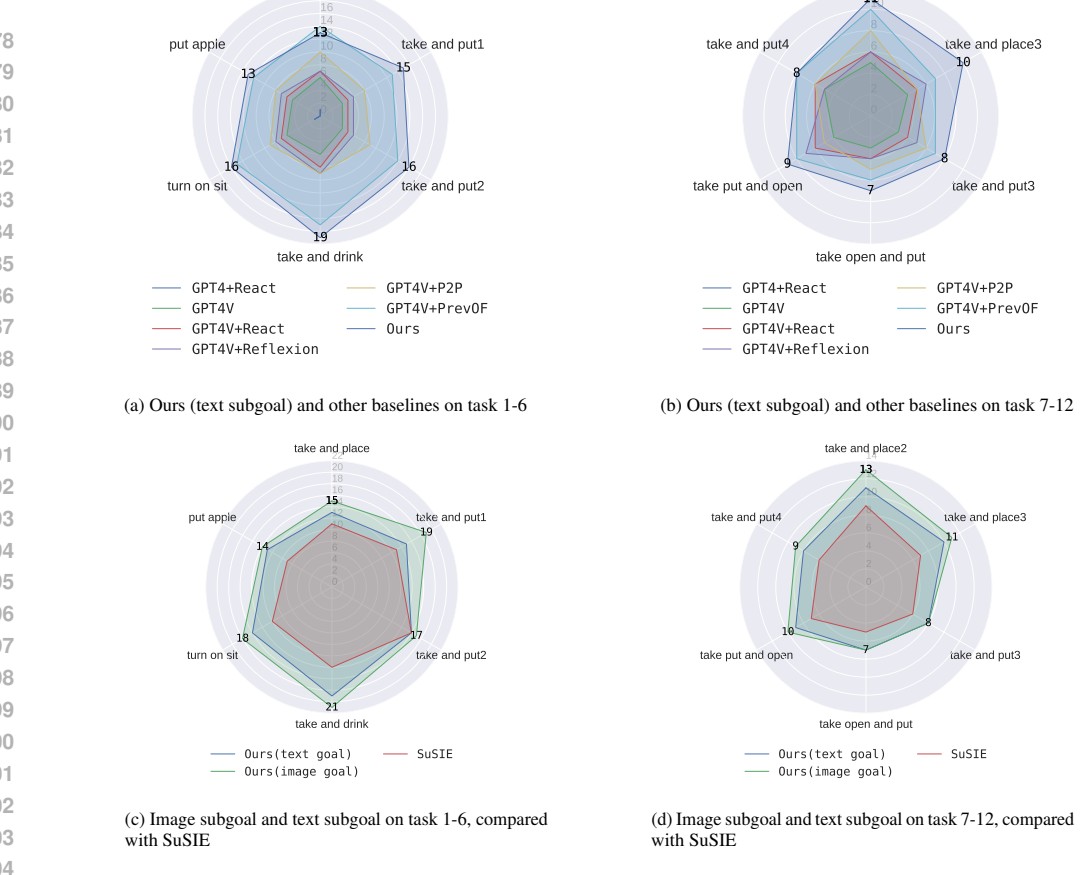

(a) Ours (text subgoal) and other baselines on task 1-6

(b) Ours (text subgoal) and other baselines on task 7-12

(c) Image subgoal and text subgoal on task 1-6, compared with SuSIE

(d) Image subgoal and text subgoal on task 7-12, compared with SuSIE

Figure 4: The success rate on 12 tasks for all the methods. Note that tasks 1-6 occur inside one room, while tasks 7-12 take place in two rooms.

accurately predicted if the prediction of the motion pattern is considered. More examples can be seen in Appendix D

## 5.2 VIRTUALHOME TASKS

**Results.** To demonstrate that our world model can well assist the LMM in task planning, we evaluate various methods on 12 tasks, each task described by an instruction, in the VirtualHome environment. Each task is tested 100 times, and the maximum step in one episode is 80. For each of the 12 tasks, we abbreviated the task names for convenience. For example, the instruction of task 1, "take the bread from the toaster and place it on the plate on the table," consists of four subtasks: a) walk to the toaster, b) grab the bread, c) walk to the plate, and d) place the bread on the plate. We use "take and place" to refer to task 1. Each task and instruction can be found in Appendix B.

These 12 instructional tasks are comprised of multiple sequential sub-tasks. For baselines, we use GPT4 combined with React (Yao et al., 2023) as the task planner and policy, denoted as **GPT4+React**, and it takes input as the JSON format text environment description. We also directly use GPT-4V to make decisions, denoted as **GPT4V**, and we also combined GPT4V with React (Yao et al., 2023) and Reflexion (Shinn et al., 2023) as the task planner and policy. When employing the Reflexion algorithm, its actor component is based on the React algorithm. These two baselines are denoted as **GPT4V+React** and **GPT4V+Reflexion**. For ablation baselines, we use the fine-tuned InstrctP2P as the world model, denoted as **GPT4V+P2P**. The world model that conditions on the previous optical flow map is denoted as **GPT4V+PrevOF**.

As shown in Figure 4, the world model significantly improves the GPT-4V ability on various long-horizon tasks. Moreover, the inclusion of optical flow information enhances the accuracy of image generation and further improves task planning performance. The results also demonstrate the effectiveness of the predicted optical flow map.

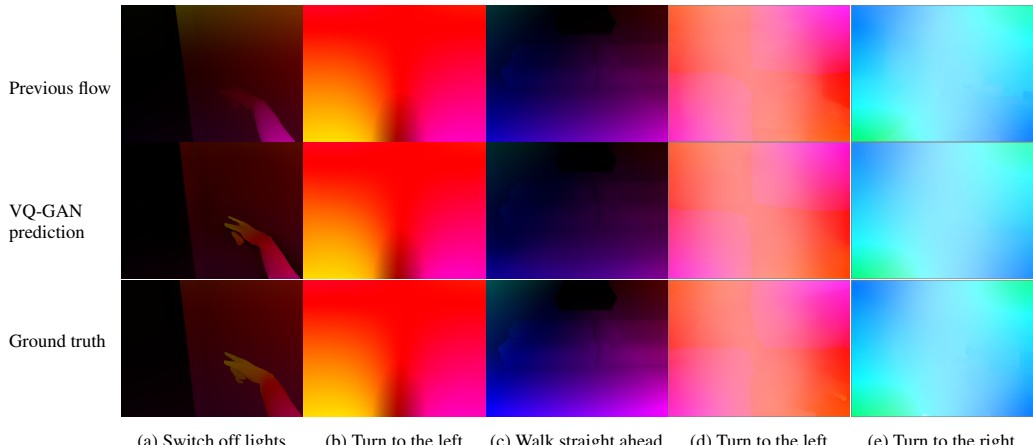

(a) Enclose the fridge  (b) Go through door  (c) Shut off the PC  (d) Take hold of pillow (e) Switch off the light  (f) Shut the stove  (g) Open the cabinet

Figure 5: Examples of the generated image subgoals. The first row is the original image, and the second row is the image subgoal generated based on the text subgoal.

(a) Switch off lights  (b) Turn to the left  (c) Walk straight ahead  (d) Turn to the left  (e) Turn to the right

Figure 6: Examples of optical flow prediction by VQ-GAN. The first 3 columns are optical flow from the VirthualHome environment. The last 2 columns are optical flow from the Habitat 2.0 environment.

**Image Subgoal *vs.* Text Subgoal.** In this part, we analyze the impact of different types of subgoals on tasks. During the goal decomposition process, the text subgoal directly outputted by the LLM task planner represents a high-level, coarse-grained description. If our method can generate images of the scene at the completion time of the subgoal, a more detailed, fine-grained description can be obtained. This might enhance the action selection ability that relies on the quality of the subgoal.

When using images as subgoals, our approach, in contrast to SuSIE (Black et al., 2023), employs a one-step planning world model to model the state images following different actions. Additionally, we utilize LMM for end-to-end pipeline of task decomposition and action selection, rather than SuSIE's goal-conditioned behavioral cloning (GCBC) for the downstream low-level policy. In Figure 4, we compare SuSIE (donated as **SuSIE**) with our method, demonstrating our method has advantages over SuSIE in long-horizon composite task planning, specially in terms of significant changes in perspective and the need for reasoning to generate subgoals.

Specifically, we have trained an InstructP2P model based on VH-1.5M to generate the image when the subgoal is completed, with the generation results illustrated in Figure 5. The decision-making results in Figure 4 show that fine-grained subgoal description is better than coarse-grained description, even if the generated image is not that accurate.

We also conduct a user study to evaluate the visual quality of the generated image-based subgoals. More details can be found in the Appendix E.

## 5.3 MOTION PATTERN

As mentioned before, we cannot obtain the optical flow from the current timestep to the next timestep. Therefore, we adopt the VQ-GAN model to predict the current optical flow map. As illustrated in Figure 6a and 6c, the quality of prediction for details is promising. Furthermore, as demonstrated in Figure 6d and 6e, the VQ-GAN trained on the VH-1.5M dataset can easily generalize to other environments. This is

Table 3: Average endpoint error (AEE) results. The lower the number, the closer the image is to the ground truth.

|  | Previous flow | Prediction flow |
|---|---|---|
| Habitat 2.0 | 3.30 | **3.09** |
| AI2-THOR | 5.00 | **4.08** |
| VirtualHome | 21.22 | **15.71** |

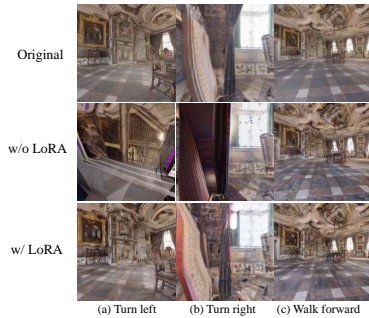

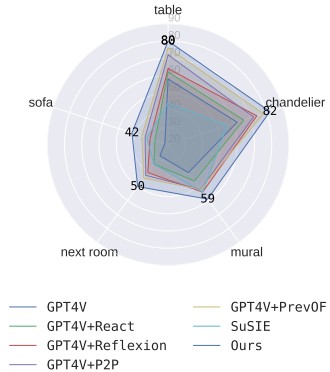

Figure 7: Examples of the generated images of the next observation in Habitat 2.0.

Figure 8: The success rate on 5 navigation tasks for all the methods in Habitat 2.0. GPT4+React is omitted due to its poor performance.

because the optical flow map is a universal feature and does not require the prediction of complex textures.

The average endpoint error (AEE) specifically measures the average distance between two motion vectors at the pixel level. As illustrated in Table 3, the gap between the predicted optical flow map and ground truth is narrower than that between the previous flow map and ground truth (current optical flow map). In addition, the model trained on VirtualHome can still predict optical flow maps in Habitat 2.0 and AI2-THOR (Kolve et al., 2017). This confirms the effectiveness and generalization of the VQ-GAN model.

### 5.4 GENERALIZATION

To assess the generalization of our method, we also evaluate its performance in a new household environment. In more detail, we choose Habitat 2.0 due to its high-fidelity scenes compared with other simulators, such as AI2-THOR. However, Habitat 2.0 does not provide any inter-frame regarding manipulation skills, which is unrealistic. Therefore, we only carry out experiments on navigation tasks.

To enhance usability, we use the pre-trained optical flow model, RAFT (Teed & Deng, 2020), to calculate the optical flow for the previous step since the optical flow cannot be directly obtained. The RAFT results are shown in the last 2 columns of Figure 6. Since VQ-GAN has demonstrated some degree of generalization ability to Habitat 2.0 in Section 5.3, we can predict the motion pattern of the new environment. The remaining task is to transfer the visual style to a new environment, and we adopt LoRA to fine-tune the world model. As shown in Figure 7, we successfully perform style transfer with a small amount of data (tens of samples), and the results with LoRA are closer to real scene images compared to those without LoRA visually.

Figure 8 shows the success rate of all methods on navigation tasks in Habitat 2.0, and we compare our method with SuSIE. We can draw the same conclusion as in the VirtualHome environment: incorporating predicted optical flow into the world model enhances the agent's decision-making capabilities. Additionally, our method achieved a high success rate, which further demonstrates its strong generalization ability. Due to the lack of generalization capability of the subgoals generated by the diffusion model in SuSIE for scenes with styles differing from the training set, the resulting subgoals lacking sufficient information, often exhibit poor quality in downstream behavior cloning methods.

## 6 CONCLUSION AND LIMITATIONS

This paper introduces EgoPlan, an embodied agent, using the LMM as the one-step planner and the Text2image model as the world model for long-horizon tasks. We demonstrate its high-quality image generation, precise optical flow prediction, and promising decision-making ability. More importantly, we have demonstrated its generalization capabilities across different environments. It is also important to acknowledge the limitations of EgoPlan. Currently, the agent uses encapsulated skills as actions. It cannot perform low-level control, *e.g.,* joint position. How to directly control low-level actions is left as future work.

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

# APPENDIX

## A   MORE RELATED WORK

### A.1   WORLD MODEL FOR DECISION-MAKING

The Dreamer series (Hafner et al., 2020; 2022; 2024) models environmental dynamics in latent space to predict future states within gaming contexts, enabling agents to learn tasks through imagination and reducing the number of interactions needed for effective learning. However, as these world models are developed in latent space rather than pixel space, they often struggle to generalize to unseen tasks and environments. A world model constructed in pixel space may offer improved generalization capabilities. Recent studies have sought to address how to learn world models from large-scale video datasets (Liu et al., 2024). In Genie (Bruce et al., 2024), researchers utilize a latent action representation, though their focus primarily revolves around 2D platform video games or simple robotic actions. By meticulously orchestrating rich data across various dimensions, UniSim (Yang et al., 2023) simulates realistic visual experiences in response to actions performed by humans, robots, and other interactive agents. Overall, the applications of world models extend beyond gaming and robotics. For instance, in Escontrela et al. (2024), frame-by-frame video prediction is employed as a mechanism for providing rewards in reinforcement learning. DynaLang (Lin et al., 2023) explores the integration of language prediction as an element of the world model, enabling the training of multimodal world models using datasets that lack explicit actions or rewards. In DynaLang, the representation is shared between vision and language within the world model.

### A.2   EMBODIED AGENT WITH LMMS

Recent methods use LMMs to assist planning and reasoning in simulation environments (Fan et al., 2022; Wang et al., 2023a; Yao et al., 2023)and robot learning (Ahn et al., 2022; Liang et al., 2023; Zeng et al., 2022). LMMs are also applied to help robot navigation (Parisi et al., 2022; Majumdar et al., 2020) and manipulation (Jiang et al., 2022; Ren et al., 2023; Khandelwal et al., 2022). Among them, ReAct (Yao et al., 2023) uses chain-of-thought prompting by generating both reasoning traces and action plans with LMMs. SayCan (Ahn et al., 2022) leverages the ability of LLMs to understand human instructions to make plans for completing tasks without finetuning LLMs. Voyager (Wang et al., 2023a) leverages GPT-4 to learn and continually discover skills during learning. While these studies demonstrate encouraging outcomes, they depend significantly on the inherent capabilities of powerful large language models (LLMs), which poses challenges for their application to smaller language and multimodal models (LMMs) with limited reasoning abilities.

The successful integration of language as a semantically rich input for interactive decision-making underscores the pivotal role of LMMs in facilitating interaction and decision-making processes (Abramson et al., 2020; Karamcheti et al., 2022; Li et al., 2022). LMMs have also been employed across various environments to support robot navigation (Parisi et al., 2022; Hong et al., 2021; Majumdar et al., 2020) and manipulation tasks (Jiang et al., 2022; Ren et al., 2023; Karamcheti et al., 2022). Recently, numerous approaches have emerged that leverage LMMs to enhance the planning and reasoning capabilities of embodied agents. For instance, SayCan (Ahn et al., 2022) evaluates the affordance of potential actions by combining their probabilities derived from LMMs with a value function. Zeng et al. (2022) integrate a language and multimodal model (LMM) with a visual-language model and a pre-trained language-conditioned policy  (Shridhar et al., 2022) to facilitate open vocabulary robotic tasks. Similarly,  Huang et al. (2022a) illustrate that LMMs can be effectively utilized for planning and executing simple household tasks, grounding LMM-generated actions by comparing their embeddings with a predefined list of acceptable actions. To incorporate environmental feedback, Inner Monologue (Huang et al., 2022b) enhances SayCan through a closed-loop principle. This principle is further employed in related works such as  (Yao et al., 2023; Huang et al., 2022b; Kim et al., 2024; Singh et al., 2023; Liang et al., 2023; Shinn et al., 2023; Wang et al., 2023c) to continuously monitor agent behaviors and refine plans accordingly for tasks in domains like computer automation and Minecraft. Furthermore, there are methods that prompt language and multimodal models (LMMs) to generate temporally abstracted actions (Zheng et al., 2023). Dasgupta et al. (2023) utilize the LMM as both a planner and a success detector for an agent, with their actor module requiring pre-training using reinforcement learning to enable the agent to adhere to natural language instructions. While these studies yield impressive results, they are heavily dependent on the inherent capabilities of

powerful LMMs, such as GPT-4 and PaLM (Chowdhery et al., 2023), which presents challenges when attempting to apply these approaches to smaller LMMs with limited reasoning abilities, such as LLaMA-7B. GLAM (Carta et al., 2023) employs RL fine-tuning to achieve functional grounding of LLMs and LMMs. However, their focus is primarily on simple primitive actions (e.g., turn left, turn right, go forward) evaluated within toy environments, such as BabyAI (Chevalier-Boisvert et al., 2018), using a significantly smaller encoder-decoder LMM, Flan-T5-780M. These primitive actions possess a similar token count and lack substantial semantic meaning, which leads to an underutilization of LMM capabilities. Consequently, they fail to adequately explore the effects of prompt design and address the imbalance within the action space, resulting in additional instability and reduced robustness.

## B  DETAILS OF VIRTUALHOME TASKS

We conducted experiments to evaluate the decision-making ability of all methods in the VirtualHome environment. In total, we investigated 12 complex tasks, with detailed instructions and reference action steps for each task as follows:

Listing 1: Instructions and subtasks.

```
<$one-house instructions$>

1. take and place: take the bread from the toaster and place it on the
    plate on the table
steps: (a). walk to the toaster
    (b). grab the bread
    (c). walk to the table
    (d). place the bread on the plate
2. take and put1: take the apple from the table and put it in the
    microwave
steps: (a). walk to the table
    (b). grab the apple
    (c). walk to the microwave
    (d). open the microwave (if the microwave is closed)
    (e). put the apple in the microwave
3. take and put2: take the book from the table and put it on the
    bookshelf
steps: (a). walk to the table
    (b). take the book
    (c). grab the book
    (d). walk to the bookshelf
    (e). put the book on the bookshelf
4. take and drink: take the water glass from the table and drink from it
steps: (a). walk to the table
    (b). take the water glass
    (c). drink the water glass
5. turn on sit: turn on the TV and sit down
steps: (a). walk to the TV
    (b). turn on the TV
    (c). walk to the chair
    (d). sit down
6. put apple: Put an apple that is on the table into the bookshelf
steps: (a). walk to the table
    (b). grab the apple
    (c). walk to the bookshelf
    (d). put the apple on the bookshelf

<$two-houses instructions$>

7. take and place2: take the frying pan from the counter and place it in
    the sink
steps: (a). walk to the counter
    (b). grab the frying pan
    (c). walk through the door
```

```
   (d). walk to the sink
   (e). place frying pan in the sink
8. take and place3: take the condiment shaker from the bookshelf and
    place it on the table
steps: (a). walk to the bookshelf
   (b). grab the condiment shaker
   (c). walk through the door
   (d). walk to the table
   (e). place condiment shaker on the table
9. take and put3: take the salmon on top of the microwave and put it in
    the fridge
steps: (a). walk to the microwave
   (b). grab the salmon
   (c). walk through the door
   (d). walk to the fridge
   (e). open the fridge (if the fridge is closed)
   (f). put salmon in the fridge
10. take open and put: take the pie on the table and warm it using the
     stove
steps: (a). walk to the table
   (b). grab the pie
   (c). walk through the door
   (d). walk to the stove
   (e). put pie on the stove
   (f). switch on the stove
11. take put and open: put the sponge in the sink and wet it by switching
     on the faucet
steps: (a). walk to the sponge
   (b). grab the sponge
   (c). walk through the door
   (d). walk to the sink
   (e). put sponge in the sink
   (f). switching on the faucet
12. take and put4: take the condiment bottle from the kitchen table and
     put it on the plate
steps: (a). walk through the door
   (b). walk to the kitchen table
   (c). grab the condiment bottle
   (d). walk to the plate
   (e). put pie on the stove
   (f). switch on the stove
```

## C    DETAILS OF VH-1.5M'S TEXT ACTIONS

The dataset includes a wide range of action sequences, each meticulously annotated with corresponding text actions. These text actions are crucial for providing contextual information that aligns visual actions with natural language descriptions. Below, we detail the process and structure used to generate the text actions for each action sequence in the dataset.

The generation of text actions for VH-1.5M involves a systematic and automated process. This process ensures consistency and variety in the text actions, which are essential for robust training and evaluation in vision-and-language tasks. The key steps in this process are as follows:

**Verb Selection:** A list of verbs related to various actions (e.g., "walk through," "close," "drink") is predefined. For each identified action sequence directory, a verb is randomly selected from the relevant list. This selection ensures a diverse representation of actions.

**Object Name Extraction:** Each directory represents the object acted upon, which signifies the object affected by the action. However, if the action does not involve an object, such as "walk through" or "turn left," no extraction is necessary.

**Phrase Construction:** Two types of phrases are constructed for each action sequence:

Next Timestep Phrase: Describes the immediate next action in the sequence. For example, "next timestep: redeposit the plate".

Goal State Phrase: Describes the intended final action or goal of the sequence. For example, "the goal state: redeposit plate".

**Prompt File Creation:** The constructed phrases are saved in a prompt json file within the respective action sequence directory. This JSON file contains two keys: "next" and "goal," corresponding to the next timestep phrase and goal state phrase, respectively.

## C.1  MORE EXAMPLES OF THE SAMPLES

We give some samples in the sequence of the task, which are shown in Figure 9, 10 and 11. Note that samples in one sequence are arranged in chronological order, with the timestep increasing from top to bottom.

## D  MORE EXAMPLES OF GENERATING IMAGES

More examples of generated images from EgoPlan can be seen in Figure 12. Each line represents a task, and the task prompts are, in order: "capture the chicken", "grasp juice", "grasp the hairproduct", "open the cabinet", "open the microwave", "go left", "make a left", "make a left-hand turn", "make a right", "turn right", "turn to the right", "walk straight ahead".

## E  USER STUDY OF SUBGOAL IMAGE GENERATION

We also conduct a user study on the image generation of the subgoal. A total of 8 users evaluated whether the generated image met the criteria of the subgoal described in the text. Each user evaluates 100 generated images for each action, and the evaluation results are shown in Table 4. The results show that most of the generated subgoal images can represent the meaning of the text subgoals. More examples of generating figures can be seen in Figure 13

Table 4: User study for the subgoal generation. The user score is the percentage of images that users consider to meet the criteria out of the total 1000 images.

|  | Close | Drink | Grab | Open | Put back | Put in |
|---|---|---|---|---|---|---|
| Mean user score(%) | 66.5 | 71.75 | 55 | 66.375 | 62.125 | 64.625 |

|  | Sit | Stand up | Switch off | Switch on | Walk through |
|---|---|---|---|---|---|
| Mean user score(%) | 79.875 | 78.75 | 73.375 | 77.875 | 79 |

## F  PROMPT OF TASK-DECOMPOSITION AND LOW-LEVEL ACTION SELECTION

We conducted experiments with detailed query prompt for each task as follows:

Listing 2: query for action selection.

```
Start working. The picture of what you can see has been given above, the
    picture is what you see from the first person perspective as the
    person in the room. Analyze the scene and all the items in the
    picture to make a task plan to complete the instruction.
The instruction is as follows:
"""
{"instruction": [INSTRUCTION]}
"""
The history is as follows:
"""
{"history": [HISTORY]}
"""
```

```
You return should follow these rules:
1. Make sure you provide 4 lines of output each time, the first line is
    the ["Preoperation"] and the secondline is the ["Postoperation"] of
    the action to be taken in the current task plan, and the third line
    is the action to be taken in the plan, which is the ["task_sequence
    "]. The fourth line is the natural language expression of the action
    taken, namely ["step_instructions"]. When output the answer, do not
    attach "step_instructions", "task_sequence", etc.
2. In addition to these, other problem such as input images is too dark
    and historical actions is empty, please DO NOT output.
3. Make sure that element of the ["step_instructions"] explains
    corresponding element of the ["task_sequence"]. That is, the fourth
    line explains the third line.
4. DO NOT USE undefined verbs. USE ONLY verbs in "HUMAN ACTION LIST".
5. The first line and the second line are detailed explanation of the
    forth line. For the task in the forth line, it must be explained in
    two parts: ["Preoperation"] and ["Postoperation"] in the first and
    second line, separately represents the action state of the agent and
    item before and after the execution of the task.
6. Look carefully at the output examples provided. DO NOT use any strings
    or spaces at the end of sentences. Never left ',' at the end of the
    sentences. STRICTLY ENSURE that the output is always four lines long,
    with no blank lines.
7. The environment given is a picture that you see from the first person
    perspective as the person in the room. Analyze the scene and all the
    items in the picture to make a task plan. If you see a picture that
    is all balck, this means there has been no task planning or execution
    before, please give a general task plan, but BE SURE to stick to the
    output format shown earlier.
8. When selecting each action for task planning, carefully think about
    the function of the action in terms of the two parts ["Preconditions
    "] and ["Postconditions"] after the action, where ["Preconditions"]
    represents the state of the environment before the action is executed
    , and ["Postconditions"] represents the state of the environment
    after the execution, after which the planning is carried out.
9. All sentences you output should NOT be double-quoted.
10. Please strictly correspond to the actions and items in the
    instructions, please strictly keep the spelling of the items, for
    multi-word items, please do not add connection symbols between words,
    for items composed of single-word, please do not split the word.
11. The history is a string that records the actions performed in the
    past few steps, separated by " ". Please plan what action to perform
    at this step based on the historical actions, instructions and the
    current picture.
12. Make sure that you output a consistent manipultation as a human. For
    example, grasping an object should not occur in successive steps.
    Consider whether the current action is simliar to the last action in
    the history. DO NOT output same two actions in row.
13. Every time you do task planning, you should consider whether the
    historical action in history and the current action have completed
    the instruction, and if so, output "Stop()" in time.
Adhere to the output format I defined above. Follow the nine rules. Think
    step by step.
```

We conducted experiments with detailed environment, role of LMM, action function, few-shot output example prompt for each task as follows:

Listing 3: prompt for environment.

```
[user]
Information about environments and objects are given as a picture that
    can be seen from the first person perspective. The picture will be
    given in the example latter.
-------------------------------------------------------
```

```
The texts above are part of the overall instruction. Do not start working
    yet:
[assistant]
Understood. I will wait for further instructions before starting to work.
```

Listing 4: prompt for role of LMM.

```
[user]
You are an excellent interpreter of human instructions for household
    tasks. Given an instruction and information about the working
    environment, you break it down into a sequence of human actions.
Please do not begin working until I say "Start working." Instead, simply
    output the message "Waiting for next input." Understood?
[assistant]
Waiting for next input.
```

Listing 5: prompt for explanation of action function.

```
[user]
Necessary and sufficient human actions are defined as follows:
"""
"HUMAN ACTION LIST"

Walk(arg1): Walks some distance towards a room or object.
Preconditions: If the environment represented by picture doesn't have the
     obj1 for the task decomposition you did to perform the action, add a
     subtask of Walk(obj1) before the task.

Grab(arg1): Grabs an object.
Preconditions: The object1 property is grabbable (except water). The
    character is close to obj1. obj1 is reachable (not inside a closed
    container). The character has at least one free hand.
Postconditions: Adds a directed edge: character holds_rh or hold_lh, obj1
    . obj1 is no longer on a surface or inside a container.

Open(arg1): Opens an object.
Preconditions: The obj1 property is IS_OPENABLE and the state is closed.
    The character is close to obj1. obj1 is reachable (not inside a
    closed container). The character has at least one free hand.
Postconditions: The obj1 state is open.

Close(arg1): Closes an object.
Preconditions: The obj1 property is IS_OPENABLE and the state is open.
    The character is close to obj1. obj1 is reachable (not inside a
    closed container). The character has at least one free hand.
Postconditions: The obj1 state is closed.

Put(arg1, arg2): Puts an object on another object.
Preconditions: The character holds_lh obj1 or character holds_rh obj1.
    The character is close to obj2.
Postconditions: Removes directed edges: character holds_lh obj1 or
    character holds_rh obj1. Adds directed edges: obj1 on obj2.

PutIn(arg1, arg2): Puts an object inside another object that is OPENABLE,
     such as stove and microwave.
Preconditions: The character holds_lh obj1 or character holds_rh obj1.
    The character is close to obj2. obj2 is not closed. If obj2 is closed
    , The character should open obj2 first and put obj1 in obj2.
Postconditions: Removes directed edges: character holds_lh obj1 or
    character holds_rh obj1. Adds directed edges: obj1 inside obj2.

SwitchOn(arg1): Turns an object on.
Preconditions: The obj1 has the property "switch." The obj1 state is off.
     The character is close to obj1.
```

```
Postconditions: The obj1 state is on.

SwitchOff(arg1): Turns an object off.
Preconditions: The obj1 has the property "switch." The obj1 state is on.
    The character is close to obj1.
Postconditions: The obj1 state is off.

Drink(arg1): Drinks from an object.
Preconditions: The obj1 property is drinkable or recipient. The character
    is close to obj1.

Sit(arg1): Sit down on an object.
Preconditions: The obj1 property is sittable. The character is close to
    obj1.

Stop(): The instruction can end the task sequence after the completion of
    the task by the planned instruction.
Preconditions: After the instruction is decomposed into a series of tasks
    , these tasks fulfill all the requirements of the instruction to be
    executed in order, that is, the instruction is completed in the
    history.
"""
-------------------------------------------------------
The texts above are part of the overall instruction. Do not start working
    yet:
[assistant]
Waiting for next input.
```

Listing 6: prompt for output example.

```
[user]
I will give you some examples of the input and the output you will
    generate.
Example 1:
"""
- Input:
The picture of what you can see has been given above.
"instruction": "take the salmon on top of the microwave and put it in the
    fridge"
"history": ""
- Output:
The microwave where the salmon is located appears to be distant or out of
    reach, and I need to approach it to interact with it.
I am now close enough to the microwave to interact with it, specifically
    to reach the salmon.
Walk(<microwave>)
Walk towards the microwave to reach the salmon on top.
"""
-------------------------------------------------------
Example 2:
"""
- Input:
The picture of what you can see has been given above.
"instruction": "take the salmon on top of the microwave and put it in the
    fridge"
"history": "Walk(<microwave>)"
- Output:
The salmon is on top of the microwave and within reach. I have at least
    one free hand to grab it.
I am now holding the salmon, which is no longer on the microwave.
Grab(<salmon>)
Grab the salmon from the top of the microwave
"""
-------------------------------------------------------
Example 3:
```

```
"""
- Input:
The picture of what you can see has been given above.
"instruction": "take the salmon on top of the microwave and put it in the
    fridge"
"history": "Walk(<microwave>)""Grab(<salmon>)"
- Output:
The fridge appears to be distant or out of reach, and I need to approach
   it to interact with it.
I am now close enough to the fridge to put the salmon inside.
Walk(<fridge>)
Walk to the fridge with the salmon
"""
-------------------------------------------------------
Example 4:
"""
- Input:
The picture of what you can see has been given above.
"instruction": "take the salmon on top of the microwave and put it in the
    fridge"
"history": "Walk(<microwave>)""Grab(<salmon>)""Walk(<fridge>)"
- Output:
Before I can put the salmon inside, the fridge must be open.
The fridge is now open, and I can place items inside.
Open(<fridge>)
Open the fridge
"""
-------------------------------------------------------
Example 5:
"""
- Input:
The picture of what you can see has been given above.
"instruction": "take the salmon on top of the microwave and put it in the
    fridge"
"history": "Walk(<microwave>)""Grab(<salmon>)""Walk(<fridge>)""Open(<
    fridge>)"
- Output:
I hold the salmon. I am close to the fridge which is now open.
The salmon is now inside the fridge, and my hands are free.
PutIn(<salmon>, <fridge>)
Put the salmon in the fridge
"""
-------------------------------------------------------
Example 6:
"""
- Input:
The picture of what you can see has been given above.
"instruction": "take the salmon on top of the microwave and put it in the
    fridge"
"history": "Walk(<microwave>)""Grab(<salmon>)""Walk(<fridge>)""Open(<
    fridge>)""PutIn(<salmon>, <fridge>)"
- Output:
After placing the salmon inside, the fridge remains open.
The fridge is now closed, securing the salmon inside.
Close(<fridge>)
Close the fridge door
"""
-------------------------------------------------------
Example 7:
"""
- Input:
The picture of what you can see has been given above.
"instruction": "take the salmon on top of the microwave and put it in the
    fridge"
```

```
"history": "Grab(<salmon>)""Walk(<fridge>)""Open(<fridge>)""PutIn(<salmon
    >, <fridge>)""Close(<fridge>)"
- Output:
I take the salmon on top of the microwave and put it in the fridge.
The instruction has been finished.
Stop()
Complete the instruction and stop the task planning
"""
--------------------------------------------------------
The texts above are part of the overall instruction. Do not start working
    yet:
[assistant]
Waiting for next input.
```

Listing 7: prompt for output format.

```
[user]
You divide the actions given in the text into detailed robot actions and
    put them together as a python dictionary.
The dictionary has three keys.
"""
- dictionary["task_cohesion"]: A dictionary containing information about
    the robot's actions that have been split up.
- dictionary["instruction_summary"]: contains a brief summary of the
    given sentence.
"""
Two keys exist in dictionary["task_cohesion"].
"""
- dictionary["task_cohesion"]["task_sequence"]: A dictionary containing
    information about the human's actions that have been split up.
- dictionary["task_cohesion"]["step_instructions"]: contains a brief text
    explaining why this step is necessary.
--------------------------------------------------------
The texts above are part of the overall instruction. Do not start working
    yet:
[assistant]
Waiting for next input.
```

## G  TRAJECTORIES OF SELF-REFLECTION IN NAVIGATION TASKS

When executing navigation tasks, the subgoal is "walk to (<somewhere>)" while the underlying actions include "walk forward", "turn left", and "turn right". Accomplishing the navigation task with such a subgoal constitutes a long-horizon composite task. In addressing these types of tasks, we employ a **React**+**Reflexion** mechanism that leverages previous actions to perform operations such as obstacle avoidance and target searching. For example, we can illustrate trajectories based on these actions.

Listing 8: Trajectory of self-reflection.

```
Trial #1
Environment: <observation image>. You are in the middle of a room.
    Looking quickly around you, you see a bed 1, a desk 2, a desk 1, a
    computer 1, a drawer 5, a drawer 4, a drawer 3, a drawer 2, a drawer
    1, a garbagecan 1, a laundryhamper 1, a cabinet 1, a shelf 6, a shelf
     5, a shelf 4, a shelf 3, a shelf 2, and a shelf 1.
Task: walk to computer 1.
> think: To solve the task, I need to find a desk, then find a computer.
> think: First I need to find a computer. From the obsersation image, a
    computer is on desk 1, from the dynamic model output, you should turn
     left to get direction.
> turn left
> success
```

```
Environment: <observation image>. You are in the middle of a room.
    Looking quickly around you, you see a bed 1, a desk
2, a desk 1, a computer 1, a drawer 5, a drawer 4, a drawer 3, a drawer
    2, a drawer 1, a garbagecan 1, a
laundryhamper 1, a cabinet 1, a shelf 6, a shelf 5, a shelf 4, a shelf 3,
    a shelf 2, and a shelf 1.
> think: First I need to find a computer. From the obsersation image, a
    computer is on desk 1, from the dynamic model output, you should turn
    left to get direction.
> turn left
> success
Environment: <observation image>. You are in the middle of a room.
    Looking quickly around you, you see a bed 1, a desk
2, a desk 1, a computer 1, a drawer 5, a drawer 4, a drawer 3, a drawer
    2, a drawer 1, a garbagecan 1, a
laundryhamper 1, a cabinet 1, a shelf 5, a shelf 4, a shelf 3, a shelf 2,
    and a shelf 1.
> think: You are get right direction of the desk, the desk is in the
    middle of the egocentric view.
> walk forward
> Nothing happens.
Environment: <observation image>. You are in the middle of a room.
    Looking quickly around you, you see a bed 1, a desk
2, a desk 1, a computer 1, a drawer 5, a drawer 4, a drawer 3, a drawer
    2, a drawer 1, a garbagecan 1, a
laundryhamper 1, a cabinet 1, a shelf 5, a shelf 4, a shelf 3, a shelf 2,
    and a shelf 1.
> think: You are get right direction of the desk, the desk is in the
    middle of the egocentric view.
> walk forward
> Nothing happens.
Environment: <observation image>. You are in the middle of a room.
    Looking quickly around you, you see a bed 1, a desk
2, a desk 1, a computer 1, a drawer 5, a drawer 4, a drawer 3, a drawer
    2, a drawer 1, a garbagecan 1, a
laundryhamper 1, a cabinet 1, a shelf 5, a shelf 4, a shelf 3, a shelf 2,
    and a shelf 1.
> think: You are get right direction of the desk, the desk is in the
    middle of the egocentric view.
> walk forward
> Nothing happens.
Status: Fail

Reflection: When we locate the computer and identify the correct
    direction, we find that the action to approach the computer is not
    responding. Observing first-person perspective images reveals that
    there is a bed in front, acting as an obstacle that prevents forward
    movement. To get closer to the computer, it is necessary to navigate
    around the bed first.

Trial #2
Environment: <observation image>. You are in the middle of a room.
    Looking quickly around you, you see a bed 1, a desk 2, a desk 1, a
    computer 1, a drawer 5, a drawer 4, a drawer 3, a drawer 2, a drawer
    1, a garbagecan 1, a laundryhamper 1, a cabinet 1, a shelf 6, a shelf
    5, a shelf 4, a shelf 3, a shelf 2, and a shelf 1.
Task: walk to computer 1.
> think: To solve the task, I need to find a desk, then find a computer.
> think: First I need to find a computer. From the obsersation image, a
    computer is on desk 1, from the dynamic model output, you should turn
    right to get around the bed.
> turn right
> success
Environment: <observation image>. You are in the middle of a room.
    Looking quickly around you, you see a bed 1, a desk 2, a desk 1, a
```

```
        computer 1, a drawer 5, a drawer 4, a drawer 3, a drawer 2, a drawer
        1, a laundryhamper 1, a cabinet 1, a shelf 6, a shelf 5, a shelf 4, a
        shelf 3, a shelf 2, and a shelf 1, a laundryhamper 2.
> think: First I need to find a computer. From the obsersation image, a
    computer is on desk 1, from the dynamic model output, you should turn
    right to get around the bed.
> turn right
> success
Environment: <observation image>. You are in the middle of a room.
    Looking quickly around you, you see a bed 1, a desk 2, a desk 1, a
    computer 1, a drawer 5, a drawer 4, a drawer 3, a drawer 2, a drawer
    1, a laundryhamper 1, a cabinet 1, a shelf 6, a shelf 5, a shelf 4, a
    shelf 3, a shelf 2, and a shelf 1, a laundryhamper 2.
> think: First I need to find a computer. From the obsersation image, a
    computer is on desk 1, from the dynamic model output, you should turn
    right to get around the bed.
> turn right
> success
Environment: <observation image>. You are in the middle of a room.
    Looking quickly around you, you see a bed 1, a desk 2, a desk 1, a
    computer 1, a drawer 5, a drawer 4, a drawer 3, a drawer 2, a drawer
    1, a laundryhamper 1, a cabinet 1, a shelf 6, a shelf 5, a shelf 4, a
    shelf 3, a shelf 2, and a shelf 1, a laundryhamper 2.
> think: First I need to find a computer. From the obsersation image, a
    computer is on desk 1, currently, there are no obstacles blocking
    your path in front of you. From the dynamic model output, you should
    walk forward to get around the bed.
> walk forward
> success
Environment: <observation image>. You are in the middle of a room.
    Looking quickly around you, you see a bed 1, a desk 2, a desk 1, a
    computer 1, a drawer 5, a drawer 4, a drawer 3, a drawer 2, a drawer
    1, a laundryhamper 1, a cabinet 1, a shelf 6, a shelf 5, a shelf 4, a
    shelf 3, a shelf 2, and a shelf 1, a laundryhamper 2.
> think: First I need to find a computer. From the obsersation image, a
    computer is on desk 1, currently, there are no obstacles blocking
    your path in front of you. From the dynamic model output, you should
    walk forward to get around the bed.
> walk forward
> success
Environment: <observation image>. You are in the middle of a room.
    Looking quickly around you, you see a bed 1, a desk 2, a desk 1, a
    computer 1, a drawer 5, a drawer 4, a drawer 3, a drawer 2, a drawer
    1, a laundryhamper 1, a cabinet 1, a shelf 6, a shelf 5, a shelf 4, a
    shelf 3, a shelf 2, and a shelf 1, a laundryhamper 2.
> think: First I need to find a computer. From the obsersation image, a
    computer is on desk 1, currently, there are no obstacles blocking
    your path in front of you. From the dynamic model output, you should
    walk forward to get around the bed.
> walk forward
> success
Environment: <observation image>. You are in the middle of a room.
    Looking quickly around you, you see a bed 1, a desk 2, a desk 1, a
    computer 1, a drawer 5, a drawer 4, a drawer 3, a drawer 2, a drawer
    1, a laundryhamper 1, a cabinet 1, a shelf 6, a shelf 5, a shelf 4, a
    shelf 3, a shelf 2, and a shelf 1, a laundryhamper 2.
> think: First I need to find a computer. From the obsersation image, a
    computer is on desk 1, currently, there are no obstacles blocking
    your path in front of you. From the dynamic model output, you should
    walk forward to get around the bed.
> walk forward
> success
Environment: <observation image>. You are in the middle of a room.
    Looking quickly around you, you see a bed 1, a desk
```

```
2, a desk 1, a computer 1, a drawer 5, a drawer 4, a drawer 3, a drawer
    2, a drawer 1, a garbagecan 1, a
laundryhamper 1, a cabinet 1, a shelf 6, a shelf 5, a shelf 4, a shelf 3,
    a shelf 2, and a shelf 1.
> think: First I need to find a computer. From the obsersation image, a
    computer is on desk 1, from the dynamic model output, you should turn
    left to get direction.
> turn left
> success
Environment: <observation image>. You are in the middle of a room.
    Looking quickly around you, you see a bed 1, a desk
2, a desk 1, a computer 1, a drawer 5, a drawer 4, a drawer 3, a drawer
    2, a drawer 1, a garbagecan 1, a
laundryhamper 1, a cabinet 1, a shelf 6, a shelf 5, a shelf 4, a shelf 3,
    a shelf 2, and a shelf 1.
> think: First I need to find a computer. From the obsersation image, a
    computer is on desk 1, from the dynamic model output, you should turn
    left to get direction.
> turn left
> success
Environment: <observation image>. You are in the middle of a room.
    Looking quickly around you, you see a bed 1, a desk
2, a desk 1, a computer 1, a drawer 5, a drawer 4, a drawer 3, a drawer
    2, a drawer 1, a garbagecan 1, a
laundryhamper 1, a cabinet 1, a shelf 6, a shelf 5, a shelf 4, a shelf 3,
    a shelf 2, and a shelf 1.
> think: First I need to find a computer. From the obsersation image, a
    computer is on desk 1, from the dynamic model output, you should turn
    left to get direction.
> turn left
> success
Environment: <observation image>. You are in the middle of a room.
    Looking quickly around you, you see a bed 1, a desk
2, a desk 1, a computer 1, a drawer 5, a drawer 4, a drawer 3, a drawer
    2, a drawer 1, a garbagecan 1, a
laundryhamper 1, a cabinet 1, a shelf 6, a shelf 5, a shelf 4, a shelf 3,
    a shelf 2, and a shelf 1.
> think: First I need to find a computer. From the obsersation image, a
    computer is on desk 1, from the dynamic model output, you should turn
    left to get direction.
> turn left
> success
Environment: <observation image>. You are in the middle of a room.
    Looking quickly around you, you see a bed 1, a desk 2, a desk 1, a
    computer 1, a drawer 5, a drawer 4, a drawer 3, a drawer 2, a drawer
    1, a laundryhamper 1, a cabinet 1, a shelf 6, a shelf 5, a shelf 4, a
    shelf 3, a shelf 2, and a shelf 1, a laundryhamper 2.
> think: First I need to find a computer. From the obsersation image, a
    computer is on desk 1. From the dynamic model output, you should walk
    forward to get around the bed.
> walk forward
> success
Environment: <observation image>. You are in the middle of a room.
    Looking quickly around you, you see a bed 1, a desk 2, a desk 1, a
    computer 1, a drawer 5, a drawer 4, a drawer 3, a drawer 2, a drawer
    1, a laundryhamper 1, a cabinet 1, a shelf 6, a shelf 5, a shelf 4, a
    shelf 3, a shelf 2, and a shelf 1, a laundryhamper 2.
> think: First I need to find a computer. From the obsersation image, a
    computer is on desk 1. From the dynamic model output, you should walk
    forward to get around the bed.
> walk forward
> success
Environment: <observation image>. You are in the middle of a room.
    Looking quickly around you, you see a bed 1, a desk 2, a desk 1, a
    computer 1, a drawer 5, a drawer 4, a drawer 3, a drawer 2, a drawer
```

```
      1, a laundryhamper 1, a cabinet 1, a shelf 6, a shelf 5, a shelf 4, a
       shelf 3, a shelf 2, and a shelf 1, a laundryhamper 2.
> think: First I need to find a computer. From the obsersation image, a
    computer is on desk 1. From the dynamic model output, you should walk
     forward to get around the bed.
> walk forward
> success
Environment: <observation image>. You are in the middle of a room.
    Looking quickly around you, you see a bed 1, a desk 2, a desk 1, a
    computer 1, a drawer 5, a drawer 4, a drawer 3, a drawer 2, a drawer
    1, a laundryhamper 1, a cabinet 1, a shelf 6, a shelf 5, a shelf 4, a
     shelf 3, a shelf 2, and a shelf 1, a laundryhamper 2.
> think: First I need to find a computer. From the obsersation image, a
    computer is on desk 1. From the dynamic model output, you should walk
     forward to get around the bed.
> walk forward
> success

Status: Success
```

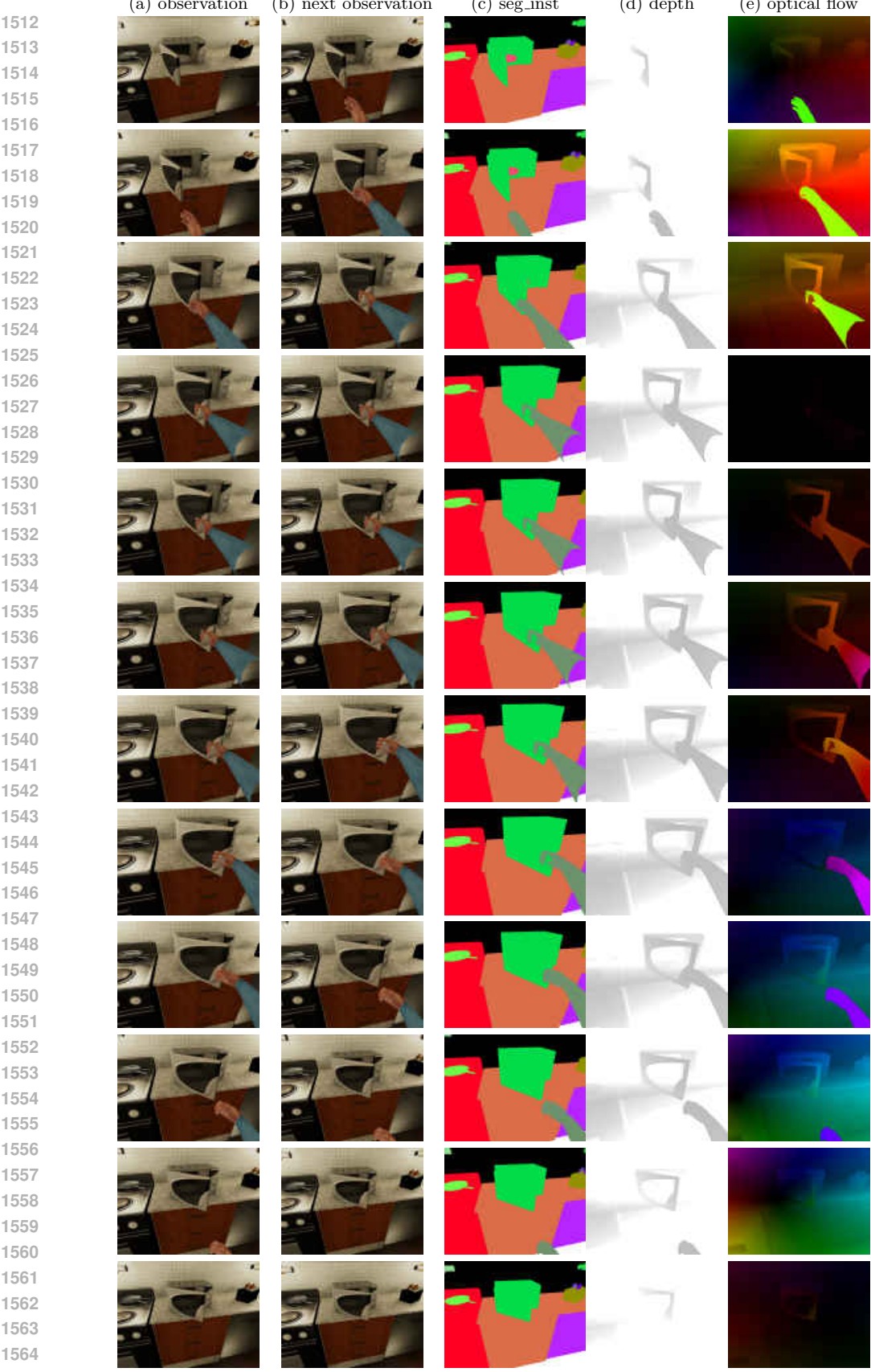

Figure 9: Samples in the sequence of closing the microwave.

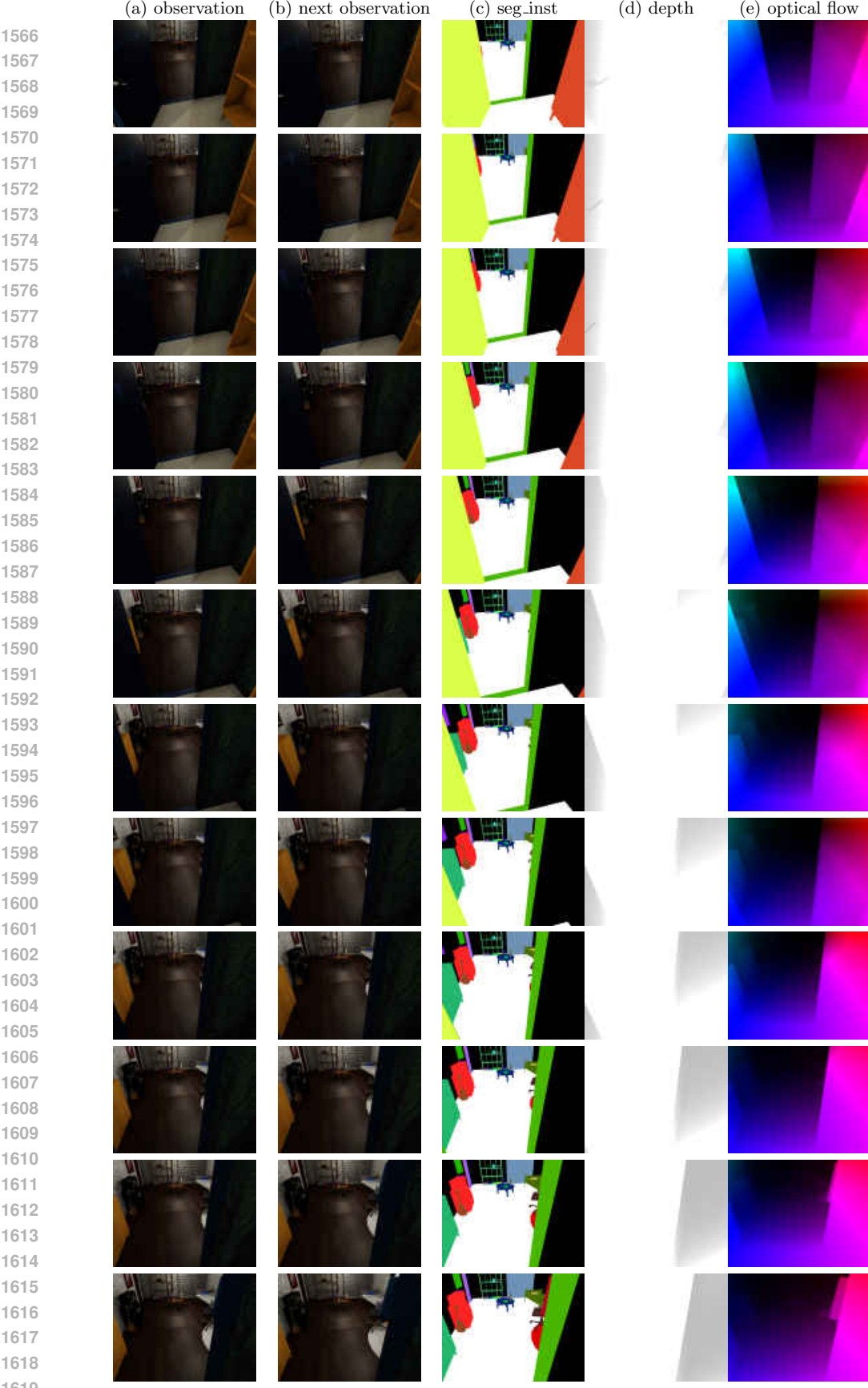

Figure 10: Samples in the sequence of walking through the door.

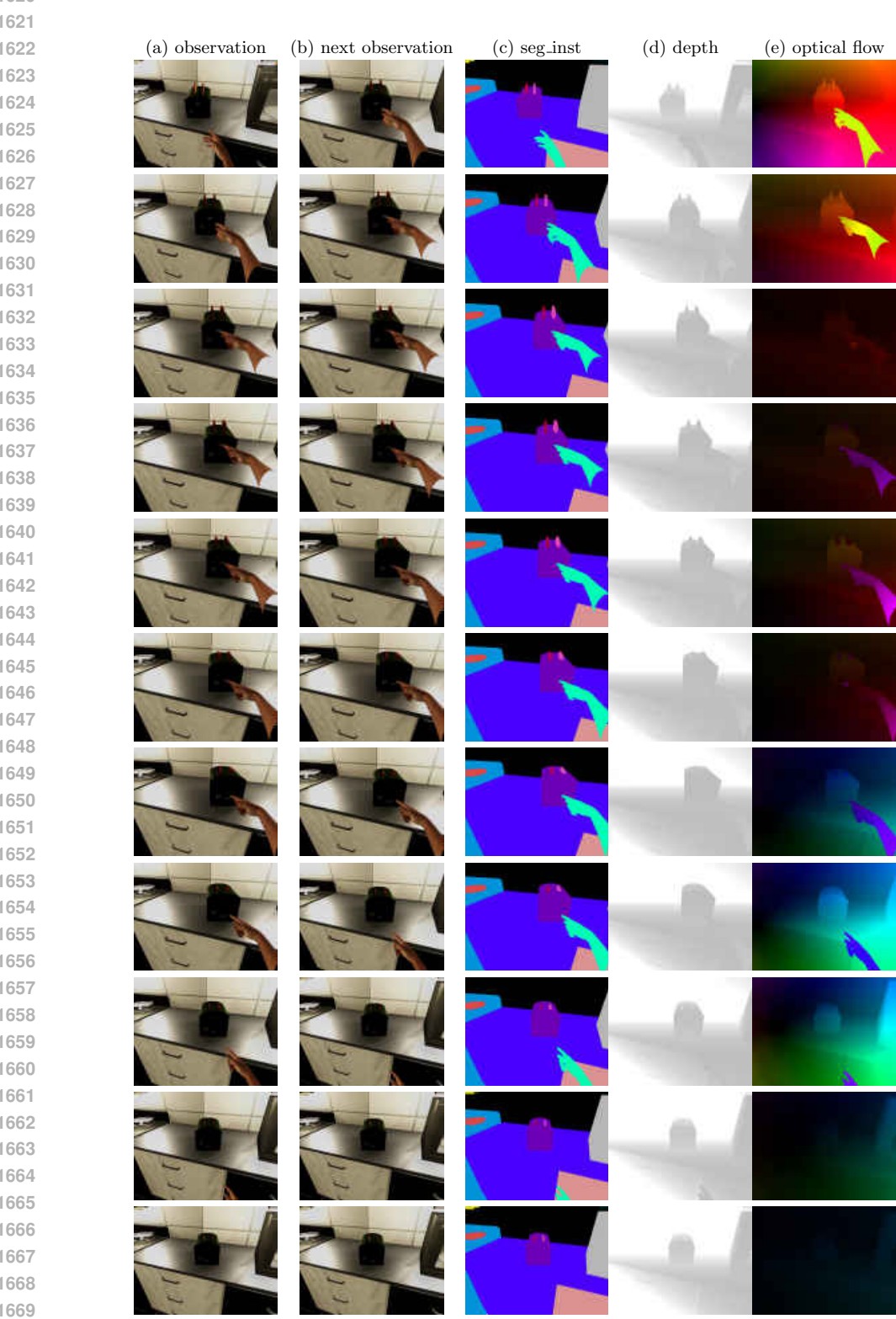

Figure 11: Samples in the sequence of switching on the toaster.

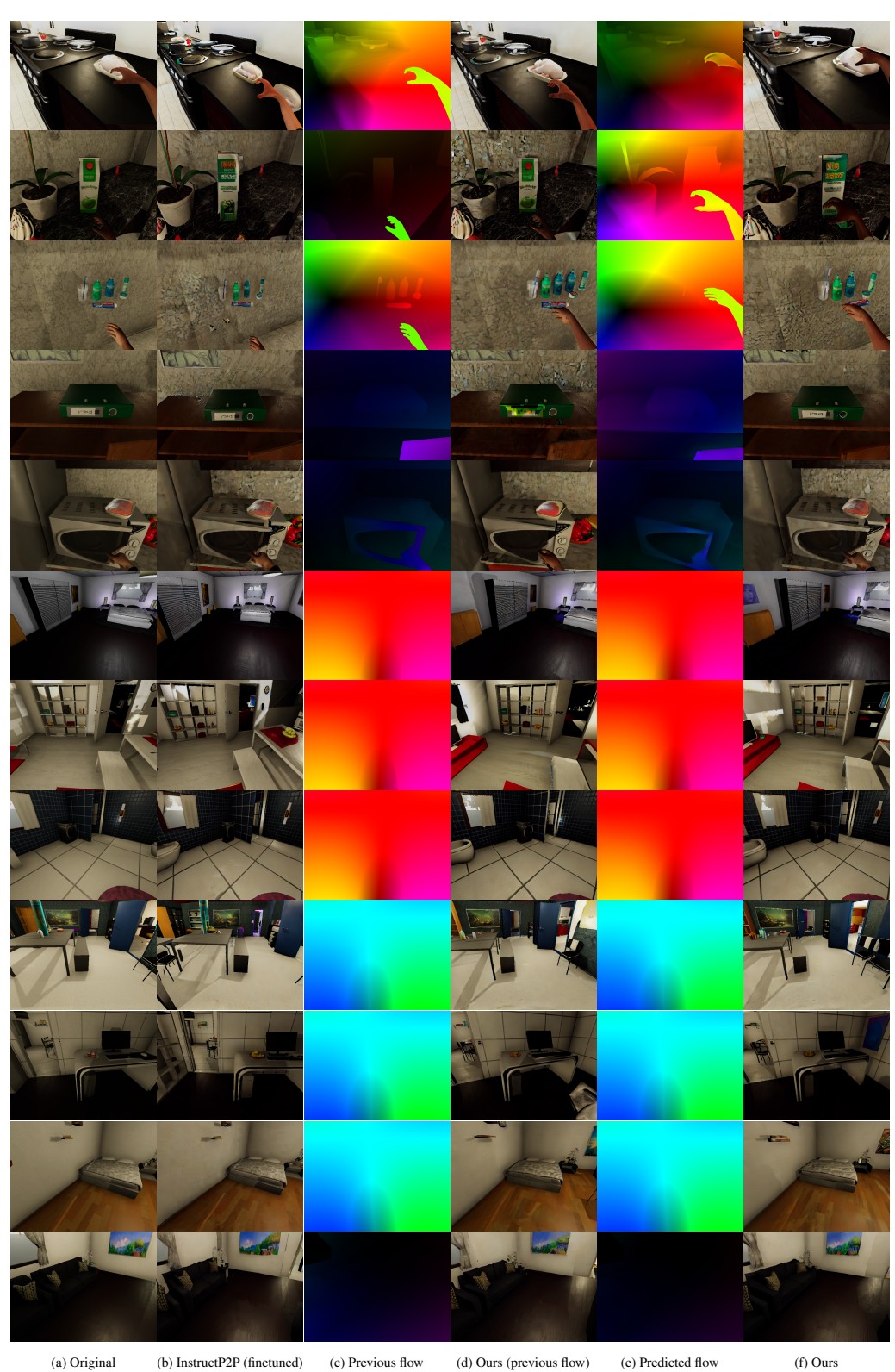

(a) Original  (b) InstructP2P (finetuned)  (c) Previous flow  (d) Ours (previous flow)  (e) Predicted flow  (f) Ours

Figure 12: Examples of the generated image of the EgoPlan in VirtualHome. We can find that in some hand reconstruction and direction understanding scenes, the model without introducing optical flow prior information often performs poorly.

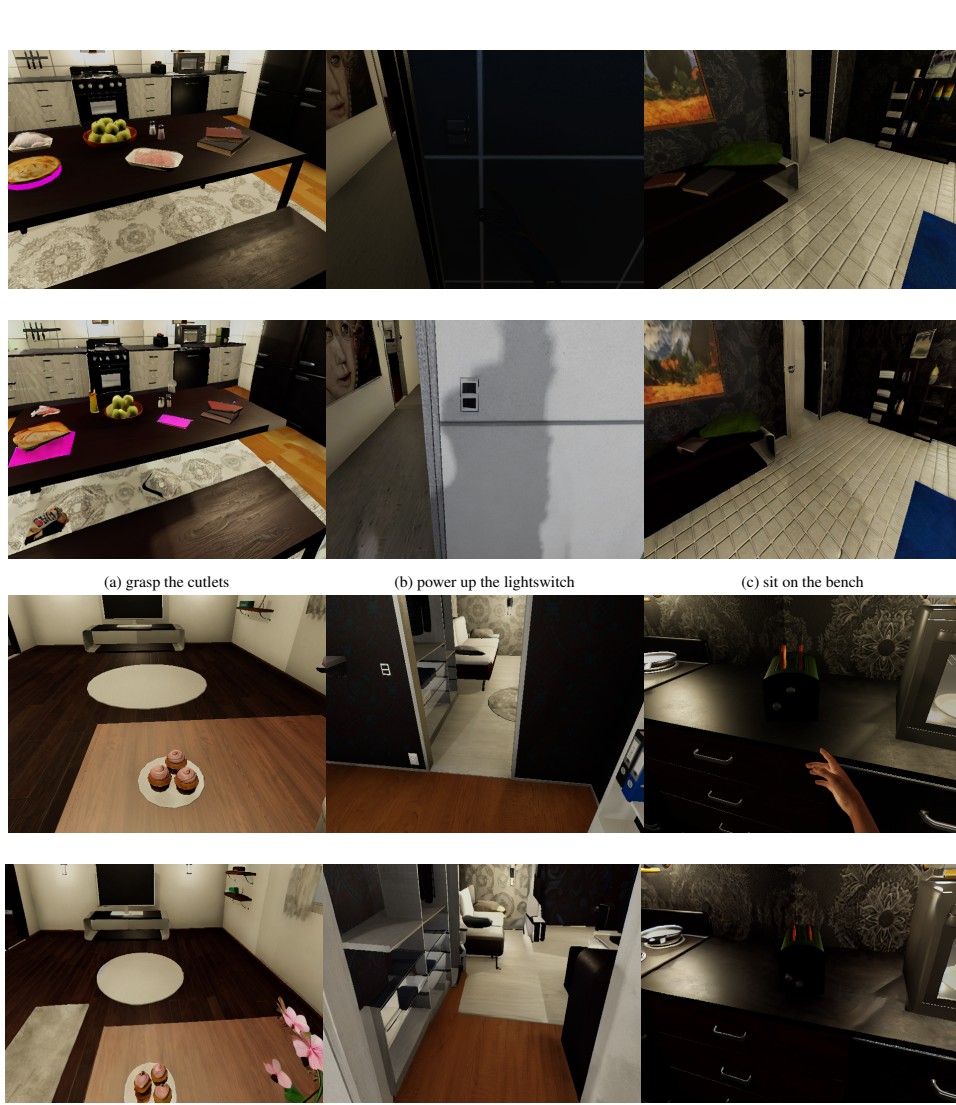

Figure 13: Examples of the generated image subgoals. The first and third rows is the original image, and the second and forth rows is the image subgoal generated based on the text subgoal.

