# OpenReview forum: "Egocentric Vision Language Planning"
_ICLR.cc/2025/Conference — ICLR 2025 Conference Withdrawn Submission_

### Official Review · Reviewer_jVJ2 · 2024-10-21

**Soundness:** 1
**Presentation:** 2
**Contribution:** 1
**Rating:** 3
**Confidence:** 5

**Summary:**

The paper introduces EgoPlan, an egocentric vision-language planning framework that leverages large multi-modal models (LMMs) and diffusion models to handle long-horizon tasks in household scenarios. EgoPlan employs a diffusion model to simulate state-action dynamics and integrates computer vision techniques like style transfer and optical flow to enhance spatial modeling and generalization across different environments. The LMM serves as a planner, decomposing instructions into sub-goals and selecting actions aligned with these sub-goals. Experiments demonstrate that EgoPlan improves task success rates compared to baselines in egocentric views.

**Strengths:**

1. Innovative Integration of LMMs and Diffusion Models: The paper presents a novel approach by combining LMMs with diffusion models for planning and action prediction in egocentric embodied environments.
2. Incorporation of Computer Vision Techniques: The use of style transfer and optical flow enhances the model’s ability to generalize across different scenes and adapt to spatial changes, which is crucial for embodied agents.
3. Dataset Contribution: The authors have collected the VH-1.5M dataset on VirtualHome, providing egocentric observations, fine-grained action information, and visualizations like optical flow, depth maps, and semantic segmentation, which can benefit future research in navigation and manipulation tasks.
4. Improved Long-Horizon Task Performance: Experimental results indicate that EgoPlan outperforms baselines in long-horizon tasks from an egocentric perspective, showcasing the effectiveness of the proposed framework.

**Weaknesses:**

1. Lack of Planning Instructions and Time Details: The paper does not provide specific planning instructions for the high-level goal decomposition shown in Fig. 2, nor does it mention the duration of the planning process. This omission makes it difficult to evaluate the efficiency and effectiveness of your planning method.
2. Insufficient Details on Diffusion Model Training: There is a lack of detailed information on how the diffusion models (particularly the World Model and the Image Subgoal Generator) were trained. Without these details, assessing the validity and reproducibility of your results is challenging.
3. Dataset Limitations and Overfitting Concerns: Relying solely on the VH-1.5M dataset may be inadequate. Additionally, there is a risk of overfitting without information on out-of-distribution (OOD) evaluations and how the training and testing data are partitioned.
4. Limited Generalizability: The applicability of the model to other scenarios, such as outdoor environments, has not been demonstrated. This raises questions about the generalization capabilities of the embodied agent in diverse environments.
5. No Discussion on Time Efficiency and System Stability: The paper lacks details on inference time efficiency and system stability, especially considering the multiple estimation components involved. Understanding potential bottlenecks is crucial for evaluating the feasibility of your method.
6. Lack of Detailed Reasoning Process: The reasoning process of the Large Language Model (LLM) is critical for evaluating and explaining outcomes, but the paper does not sufficiently discuss this aspect. It seems the dataset contains only subgoal text without detailed reasoning steps.
7. Confusion in Fig. 7 Results: In Fig. 7, the results without LoRA fine-tuning sometimes appear better than those with LoRA. This raises concerns about potential overfitting and the model’s ability to handle significant environmental changes.

**Questions:**

1. What Is the Duration of the Planning Process?
Could you provide more details on the time taken for the high-level planning process in Figure 2 and the specific planning instructions used?
2. Training Method for the Image Subgoal Generator:
How did you train the diffusion model used for subgoal prediction? Is it conditioned only on the final goal? When there is a significant difference between the subgoal and the initial screen, how do you ensure accurate outputs without environmental context? If the output quality is consistently high, is there a risk of overfitting?
3. Details on Dataset Partitioning and Evaluation:
How did you partition the dataset for training and evaluating the two diffusion models? How do you maintain output quality when the predicted image differs greatly from the input? Did you perform out-of-distribution (OOD) evaluations to address potential overfitting issues?
4. Assessment of Generalizability:
Have you tested your model in other scenarios, such as outdoor environments, to evaluate its generalizability and potential applicability?
5. Inference Time Efficiency and System Bottlenecks:
Could you provide more information on the system's inference time efficiency? Which components might be potential bottlenecks? Have you conducted ablation studies to assess the system’s stability?
6. Inclusion of LLM Reasoning Process in the Dataset:
Does your dataset include detailed LLM reasoning processes, or only subgoal texts? Could you provide some comparative reasoning cases with multi-modal input to offer deeper insights?
7. Clarification on Fig. 7 Results:
Could you explain why, in Fig. 7, the results without LoRA fine-tuning sometimes appear better than those with LoRA? Does this indicate potential overfitting or limitations in the model’s ability to handle significant environmental changes?

---

### Official Review · Reviewer_y3Gf · 2024-11-01

**Soundness:** 2
**Presentation:** 2
**Contribution:** 2
**Rating:** 3
**Confidence:** 4

**Summary:**

The authors leverage LLM/VLMs and T2I models to construct an embodied planning pipeline capable of one-step planning. The model is tested on VirtualHome and compared to various baselines. A new virtual-home based dataset is collected.

**Strengths:**

1. Authors perform a number of ablation studies to demonstrate the usefulness of each model component
2. Authors compare their model to many different baselines.

**Weaknesses:**

1. The authors' use of "world model" to describe the paper's diffusion (image editing) component is highly exaggerated. By definition, world models should record and keep track of complete and accurate environment states. However, here the diffusion model is merely LoRA finetuned to edit the provided image in an in-distribution manner. The authors also fail to explain why their diffusion module can remotely constitute a world model in their methods section.

2. The InstructP2P model is known to be not very strong with physical understanding (when asked to generate the new scene after a significant action / significant view shift has taken place, it often fails). If the authors are able to overcome this issue, more visual examples should be demonstrated on harder cases.

3. The model is only able to perform greedy one-step planning, which means it has no way of optimizing its actions based on global goal. While world models are sometimes used to solve this problem, in this work the provided "world model" module seems far from being able to support this.

4. Although relatively easy to set-up, the VirtualHome simulator is quite old and visual simulation quality is not very good compared to newer simulators (eg. Behavior, Robosuite/Robocasa, etc.) or real world robotic datasets (eg. DROID). Experiments on VirtualHome alone is not a good-enough indicator whether or not the model can be adapted to real-world circumstances.

5. Some typos and incoherent sentences: for example "Introduce optical flow into the world model leads the world model more sensitive to action position changes and adapt to scene changes during navigation." (L92-93)

**Questions:**

1. Would it be beneficial to better define the task in your dataset using a formal markov decision process?
2. The main results figure seems to not display baseline performance numbers, where are they?

---

### Official Review · Reviewer_xwpK · 2024-11-04

**Soundness:** 1
**Presentation:** 3
**Contribution:** 2
**Rating:** 3
**Confidence:** 4

**Summary:**

This paper proposes EgoPlan, which casts a diffusion model as world model and an LLM as a high-level planner. Specifically, the diffusion model synthesizes future scenes to corresponding to several admissible actions, while the LLM predicts the next action based on the most probable future scene. The authors also propose a new dataset, dubbed as VH-1.5M, which annotates the segmentation map, depth map, and the optical flow for the trajectories collected from VirtualHome. They conduct experiments on various tasks in the VirtualHome environment. They also evaluate the quality of the generated images and optical flows on several different datasets.

**Strengths:**

1. The paper is well organized and easy to follow.
2. The idea of using optical flow to generalize across environments is reasonable and novel.

**Weaknesses:**

1. The idea of using generative model as world model [1,2,3,4] and LLM as task planner [5,6] have been widely studied in previous works.
2. (contd. 1.) The unique contribution of this paper appears to be the use of optical flow to generalize the world model across diverse environments. However, the experiment results are not sufficient to support this claim. Including task execution results rather than solely optical flow error across different simulators, could provide more comprehensive evidence and improve the robustness of the findings.
3. For the main experiment (Figure 4), presenting the results in a table rather than a figure could enhance clarity. It is unclear to me how the world model benefits the final task execution compared to directly employing GPT-4V for task planning.
4. The authors are encouraged to evaluate their methods on more challenging tasks that require long-term planning capabilities, such as ALFRED or RxR-Habitat, to further validate their approach.

Overall, while the paper presents an interesting direction, my main concern is that additional foundational experiments would strengthen its claims. The authors are encouraged to consider these comments to enhance paper’s contributions.

[1] Contrastive Learning as Goal-Conditioned Reinforcement Learning. NeurIPS 2022.

[2] Mastering Atari with Discrete World Models. ICLR 2021

[3] Learning Latent Dynamics for Planning from Pixels. ICML 2019.

[4] Dream to Control: Learning Behaviors by Latent Imagination. ICLR 2020.

[5] LLM-Planner: Few-Shot Grounded Planning for Embodied Agents with Large Language Models. ICCV 2023.

[6] Do As I Can, Not As I Say: Grounding Language in Robotic Affordances. CoRL 2022.

**Questions:**

Please see my weakness section.

---

### Official Review · Reviewer_AFuU · 2024-11-05

**Soundness:** 3
**Presentation:** 3
**Contribution:** 3
**Rating:** 6
**Confidence:** 4

**Summary:**

This work has collected a dataset on Virtualhome viewing an action of the agent as a trajectory, with egocentric information.  The EgoPlan framework is introduced combining LMM for planning and a diffusion world model for dynamics prediction. Optical flow modality is used for advancement. The framework demonstrates improved performances on generation quality, VirtualHome and Habitat.

**Strengths:**

1. The framework is well-motivated and reasonable.
2. The data effort will be of good use to future works.
3. The paper is well-organized and easy to read.
4. The proposed method outperforms the baseline.

**Weaknesses:**

1. Some crucial ablation studies are missing. How does the framework perform without optical flow and style transfer?
2. Some related works may share similar motivations using diffusion models for world dynamics, and dynamics for planning, you may consider to cite.

[1] 3D-VLA: A 3D Vision-Language-Action Generative World Model
[2] Diffusion Reward: Learning Rewards via Conditional Video Diffusion

**Questions:**

See weakness part

---

### Note · Authors · 2024-11-14

I have read and agree with the venue's withdrawal policy on behalf of myself and my co-authors.